# Targeting the JAK2/STAT3 Pathway—Can We Compare It to the Two Faces of the God Janus?

**DOI:** 10.3390/ijms21218261

**Published:** 2020-11-04

**Authors:** Anna Jaśkiewicz, Tomasz Domoradzki, Beata Pająk

**Affiliations:** Independent Laboratory of Genetic and Molecular Biology, Kaczkowski Military Institute of Hygiene and Epidemiology, Kozielska 4, 01-163 Warsaw, Poland; domoradzki.tomasz@gmail.com

**Keywords:** muscle cachexia, targeted therapy, pSTAT3 inhibitors

## Abstract

Muscle cachexia is one of the most critical unmet medical needs. Identifying the molecular background of cancer-induced muscle loss revealed a promising possibility of new therapeutic targets and new drug development. In this review, we will define the signal transducer and activator of transcription 3 (STAT3) protein’s role in the tumor formation process and summarize the role of STAT3 in skeletal muscle cachexia. Finally, we will discuss a vast therapeutic potential for the STAT3-inhibiting single-agent treatment innovation that, as the desired outcome, could block tumor growth and generally prevent muscle cachexia.

## 1. Muscle Cachexia

Muscle cachexia is one of the most common and severe symptoms of advanced cancer, often observed in the course of upper gastrointestinal tract cancers (especially pancreatic, stomach, and esophageal cancers), head and neck cancers, lung cancer, and non-Hodgkin’s lymphomas [1,2]. As defined by the Cachexia Consensus Conference in 2008, cachexia is “metabolic syndrome associated with the underlying disease and characterized by muscle loss with or without fat loss.” The complex molecular mechanisms underlying the gradual reduction of body tissue mass have not been fully understood. Cancer cachexia is also often referred to as cachexia–anorexia syndrome. Anorexia in cancer patients is associated with the predominance of signals suppressing appetite in the hypothalamus—proopiomelanocortin and anorexigenic action of pro-inflammatory cytokines: IL-1α, IL-1β, IL-6, TNF-α. Additionally, the effect is exacerbated by significant metabolic changes, such as energy expenditure at rest and disturbed metabolism of carbohydrates, proteins, and lipids [3]. However, the mechanisms of losing muscle mass in cancer cachexia have a different background than starvation. Oncological patients have reduced body weight due to the gradual decrease in muscle and fat mass, while non-muscle proteins are preserved [3]. Several studies showed that depending on the type of cancer, loss of muscle mass affects 30 to 80% of patients and is responsible for a drastic reduction in quality of life, as well as reducing the effectiveness of chemotherapy, often being the direct cause of death [4,5]. Among factors causing cachexia, the leading role is attributed to substances with cachectic activity produced by cancer cells and the immune system, mainly cytokines, including the vital IL-6 and others, such as TNF-α, IL-1, IFN-γ, lipolysis activating factor (LMF), and proteolysis inducing factor (PIF) [6,7]. Furthermore, skeletal muscle proteins’ degradation processes via lysosomal pathways and ubiquitin–proteasome systems play an essential role in muscle atrophy and are overactive in over 50% of cancer patients [8]. Rapidly progressive cancer cachexia syndrome leads to multi-directional changes, affecting all aspect of patients’ wellness, including anemia, nutritional deficiencies, loss of muscle mass and activity limitation, impairment of internal organs and immune system function, changes in external appearance, depression, weakening social bonds, deterioration of quality of life and, as a consequence, faster death of the patient [9]. Because weight loss is an important prognostic factor in cancer patients, the inability to stop cancer progression of cancer cachexia is often a critical, ultimately determining factor in terminating chemotherapy treatment due to the organism’s poor condition.

While significant development of molecular biology, treatment strategies, and novel drugs dedicated to treating several oncological diseases has been introduced, unfortunately, there is still no significant progress in pancreatic cancer therapy, in almost all cases, associated with muscle cachexia. Moreover, muscular cachexia is still wholly deprived of the possibility of pharmacological intervention, and the only recommendation for the patient is to use a high-protein diet. Implementing an appropriate diet is very often difficult to achieve since one of the paraneoplastic syndromes is appetite suppression [10]. Thus, the consumption of recommended high amounts of protein through the diet itself, which could support maintaining muscle mass, is impossible for most patients. In advanced cases, enteral nutrition has to be implemented. Hitherto, new drug candidate clinical trials have been associated with the administration of progesterone derivatives—medroxyprogesterone [4], megestrol acetate [11], ghrelin [12], and delta-9-tetrahydrocannabinol [13]—as appetite stimulators and weight loss limiting agents, as well as corticosteroids, erythropoietin, and angiotensin converting enzyme (ACE) inhibitors as muscle tissue metabolism modulators [14,15]. Unfortunately, stimulation of the hunger and satiety center and increased food intake are insufficient to compensate catabolic processes intensified in cancer cachexia and cannot reconstruct or even inhibit muscle mass loss [16]. In turn, a long-term treatment using anabolic hormones is not possible due to the strong immunosuppressive effects, limiting anticancer therapy effectiveness. Recent clinical trials in cachexia therapy are also quite limited and are mainly focused on dietary supplements limiting oxidative damage and protein loss in the skeletal muscles. A summary of clinical trials targeting cachexia is shown in Table 1.

## 2. STAT3 Protein

It seems that the search for effective treatment and new drug candidates requires the knowledge of the exact molecular mechanisms that determine the development of cachexia–anorexia syndrome, with particular emphasis on the role of endogenous factors produced by proliferating cancer cells. Targeting molecular pathways involved in oncogenesis, whose excessive activity is responsible for skeletal muscle degradation, is a promising strategy. A great example of such an approach is JAK/STAT signaling pathway inhibition, which, on the one hand, could stop the expansion of certain types of tumors and, on the other, has a potentially protective effect on skeletal muscles. Among cellular proteins, particular emphasis should be put on the signal transducer and activator of transcription 3 (STAT3), which drives cancer progression and is also crucial in muscle cachexia.

## 3. STAT3 Protein in Oncogenesis

Constitutive activation of STAT3 leading to cancer is the result of increased cytokine production (IL-6, IL-10), continuous cytokine receptor activation (VEGFR/EGFR), or non-receptor tyrosine kinases (JAKs, Src, Abl) [17]. Activation of JAK, MAPK, or mTOR kinase results in phosphorylation of tyrosine or serine residues in the C-terminal domain of the STAT3 protein and its dimerization. The active STAT3 dimer moves to the cell nucleus to initiate transcription of target genes [18]. In tumor cells, STAT3 is constitutively active. Constitutive activation of STAT3 has been described in hematopoietic disorders such as myeloma, Hodgkin lymphoma, anaplastic large cell lymphoma (ALK), angioimmunoblastic T cell lymphoma, adult T cell lymphoma/leukemia, and mantle lymphoma [19,20].

Moreover, continuous cytokine-mediated STAT3 stimulation has been reported in many cases of solid tumors—head and neck cancer, melanoma, prostate, breast, colon, and gliomas [20,21]. In cancer cells, constitutive activation of STAT3 is necessary for promoting the overexpression of genes that encode anti-apoptotic proteins and that are regulators of cell cycle and angiogenic factors [22]. Strong STAT3 expression in cancer cells results from the loss of inhibitory signals and the predominance of factors causing its continuous activation. The result of STAT3 activity is the accumulation of IL-6/JAKs, EGRF, Src, transcription factors, and oncoproteins, induced by inhibition or deletion of negative regulatory proteins or over-stimulation of intracellular and extracellular factors [23]. Due to the various stimuli mentioned above regarding STAT3 overexpression, universal upstream-targeted therapy is not possible. Similar sustained STAT3 activity may be determined by the inactivation of STAT3 suppressors such as SOCS (suppressor of cytokine signaling) and PIAS (protein inhibitor of activated STATs). Protein tyrosine phosphatases may also cause similar results as activation of upstream STAT3 signaling pathways [24]. Increased activation of STAT3 in cancer cells results in its continuous presence in the cell nucleus and gene expression disorder. Many studies indicate that STAT3 participates in regulating critical processes for the development and progression of cancer. It plays a role in cell survival, proliferation, angiogenesis, metastasis, and cell protection against the body’s immune response [21] (Figure 1).

## 4. STAT3 Protein in Skeletal Muscle

Skeletal muscles are considered one of the most malleable body tissues, whose functioning, mass, and metabolism are shaped by both endo- and exogenous stimuli, guaranteeing a balance between anabolic and catabolic processes, and thus muscle homeostasis. In the skeletal muscle cells, the basic range of STAT3 activity is necessary to develop skeletal muscle satellite cells properly. In vitro studies on the knock-out of the gene encoding the STAT3 protein reduces the expression of muscle differentiation’s primary markers, such as MyoD and myogenin [27]. Similar observations are provided by in vivo studies in which STAT3 deletion causes impairment of post-traumatic muscle regeneration [27]. In contrast, transient use of the STAT3 inhibitor in mice accelerates the repair process of injured muscle tissue [27]. In vivo studies have also shown that in muscle injury, STAT3 is responsible for the renewal of satellite cells [28].

On the other hand, in vivo studies on conventional STAT3 knockout mice indicates early embryonic lethality at E6.5 to E7.5 levels, resulting in rapid embryo degeneration between E6.5 and E7.5 without apparent mesoderm formation [29]. STAT3 ablation results in embryonic death less than one day before embryo development in functional cardiomyocytes at E7.5 to E8.5 [29]. It seems likely that STAT3 is located in the embryo at the cardiac field formation site, suggesting STAT3 contribution to cardiomyogenesis. The role of STAT3 in embryogenesis remains unelucidated because STAT3 knockout is fatal to embryos by inhibiting cardiomyocyte formation [29]. The above reports illustrate many faces of STAT3, the presence of which seems to be of crucial importance for muscle cells.

STAT3 is the most crucial element of the interleukin 6 (IL-6) and JAK2 signaling pathway, regulating skeletal muscle mass, growth, repair, and regeneration [30]. Numerous studies indicate the critical role of STAT3 activity in skeletal muscle pathology development like muscle cachexia [31].

Activation of the STAT3 pathway has been shown to induce muscle tissue atrophy in Duchenne muscular dystrophy (DMD), Merosin-negative congenital muscular dystrophy (MDC1A), sepsis, and in the vast majority of cancers [32,33,34]. STAT3 also affects the nervous and cardiovascular systems, directly affecting skeletal muscle function [35,36,37,38]. The vast majority of papers suggest the participation of STAT3 in both physiological and pathological processes, demonstrating the multidimensional nature of STAT3 in skeletal muscle. STAT3 activity is a necessary condition during muscle tissue development and maintains its homeostasis, while STAT3 inhibitors seem to be a promising element in diseases involving muscle wasting. The IL-6/JAK2/STAT3 signaling pathway is currently the central object of research on cancer cachexia. In vivo studies show a pro-catabolic effect of STAT3 on skeletal muscles in experimentally induced cachexia [39]. Permanent activation of the acute phase protein response is considered the primary molecular mechanism contributing to cancer cachexia. The IL-6/STAT3 signaling pathway induces a loss of muscle mass in experimental cachexia models via two pathways, where on the one hand, STAT3 phosphorylation leads to activation of hepatic acute phase protein gene expression [40].

On the other hand, skeletal muscles are a primary source of proteins triggering acute phase reaction [41]. The molecular mechanism of muscle mass loss via increased STAT3 activity activates the ubiquitin–proteasome system, either directly by STAT3 binding to ubiquitin-related gene promoters or indirectly by activation of caspase-3 [39]. An additional factor appears to be a reduction in protein synthesis by inhibition of mTOR activity by AMP-activated kinase (5′AMP-activated protein kinase, AMPK), observed mainly during the terminal stages of cancer cachexia [42]. Skeletal muscles are subjected to damaging stimuli, primarily due to cytokines produced by tumor cells, such as IL-6, TNF-α, and IFNγ, responsible for inflammation. The inflammatory response has been shown to promote cancer cell proliferation while inhibiting skeletal satellite cell differentiation. Satellite skeletal muscle cells maturing in a state of cachexia are characterized by Pax7 expression, corresponding with their proliferation, with no expression of myogenin, which proves impaired regeneration ability [43]. Thus, it is suspected that STAT3 promotes the expansion of the pool of satellite cells and plays a negative role in the differentiation and repair processes of skeletal muscles [44]. On the other hand, some of the most recent reports on the role of STAT3 in the functioning of skeletal muscle mitochondria are surprising and confirm the need for further research on STAT3 in skeletal muscles. Using STAT3 knockout mice proved that the loss of STAT3 does not affect the mitochondrial and physiological functions of skeletal muscles, both in in vivo and ex vivo studies [45].

As mentioned above, skeletal muscle is one of the most dynamic tissues of the body, maintaining an appropriate balance between anabolic and catabolic processes. Many growth factors, cytokines, and myokines produced by skeletal muscle cells play a vital role in the local regulation of inflammation and skeletal muscle regeneration in various pathological states. The vast majority of the activity of STAT3 depends on the IL-6/JAK2/STAT3 signaling pathway, which probably explains the contradictory reports on STAT3 activity in skeletal muscles. IL-6 is a pleiotropic cytokine released in large amounts during infection, autoimmunity, and cancer [30]. Low IL-6 levels may promote satellite cell activation and myotube regeneration, while chronically elevated production of that myokine promotes skeletal wasting [30].

## 5. STAT3 Protein in Cancer Cachexia

Cachexia is a multifactorial syndrome of skeletal muscle and fat loss, resulting in progressive weight loss closely associated with the overall cancer survival prognosis. Research emphasizes the critical role of humoral factors secreted by cancer cells in the patient’s tissues in the regulation of processes leading to cachexia, which also include pro-inflammatory cytokines IL-6, TNF-α, IFNγ, IL-1α, and IL-1β. By the 90s of the last century, it was known that IL-6 is one of the main factors stimulating pathological changes in muscles. It was shown that using anti-IL-6R antibodies can inhibit muscular atrophy in IL-6 overexpressing transgenic mice [46]. In studies using ApcMin/+ mice, an established colon cancer and a cachexia model, administration of IL-6 receptor-specific antibodies prevented weight loss and suppressed protein degradation without affecting muscle protein synthesis or IGF-1-associated signaling [42]. In vitro studies in mouse myotubes have shown that IL-6 reduces the half-life of long-lived proteins by increased activity of the 26S proteasome and cathepsins B and L. This observation suggests that IL-6 increases the degradation of proteins in the muscles by activating both non-lysosomal (ubiquitin–proteasome) and lysosomal (cathepsin) processes. As research has shown [47], in addition to IL-6, IFNγ plays a central role in cachexia. In vivo studies, passive immunization against IFNγ and TNF-α of tumor-bearing rats allowed the intake of food, weight preservation, longer life, and better tolerance of larger tumors than in rats receiving a control antibody. Besides, IFNγ has been shown to inhibit MyHCII expression in skeletal muscle, an essential protein for skeletal muscle cell degradation [48]. Recent research by Ma et al. [49] proves that regardless of the interaction among IL-6, IFNγ, and TNF-α stimulation, the muscle mass loss is activated by the pSTAT3/NF-κβ pathway.

## 6. Synthetic STAT3 Inhibitors in Anticancer Therapy

The number of STAT3-interacting peptides currently tested in preclinical studies is almost countless. Peptides such as PY*LKTK [50], Y*LPQTV [51], SS610 [52], S3I-M2001 [53], STA-21 [54], and S3I-201 [54] bind to the SH2 domain and thus inhibit STAT3 dimerization. In turn, static [55] G-quartet oligodeoxynucleotides (ODN) [56] binding to the SH2 domain of STAT3 or JSI-124, and withacnistin that binds to JAK2 are examples of molecules that block the proper phosphorylation of STAT3. Another group of STAT3 inhibitors represented by IS3 295 [57], CPA-1, CPA-7 [58], galiellalactone [57], and peptide aptamers [59,60], bind to the DNA-binding domain (DBD) responsible for the binding of STAT3 to DNA, thereby blocking the transcriptional activation of STAT3-targeted genes. Regardless of whether the peptides mentioned above inhibit phosphorylation, dimerization, or binding of STAT3 to DNA, the effect of their interaction is the inhibition of malignant cell growth and transformation, intensification of apoptosis, and reduction of the invasiveness of cancer cells [61], which means that they have enormous potential in oncological research.

WP1066 is one of the novel pSTAT3 inhibitors that is currently registered as an “orphan drug” in the US Food and Drug Administration (FDA) and has received approval for conducting a phase I clinical trial in the treatment of glioblastoma (GBM) and melanoma metastases to the brain [62]. Due to its high bioavailability, after oral administration, WP1066 binds specifically both to JAK2 and JAK2 V617F—the mutated form characteristic for cancer cells, thus inhibiting phosphorylation and kinase activation [63]. Numerous in vitro and in vivo studies on renal cell carcinoma [64], bladder cancer cells [65], and GBM [66,67] proved the high effectiveness of WP1066 in limiting the growth and survival of cancer cells. Studies on tumor metastasis suggest a positive role of WP1066 in reducing breast cancer cells’ metastasis to the brain [68]. It should be emphasized that the high activity of STAT3 is also one of the main mechanisms of pancreatic cancer cell proliferation [69]. The fourth, most common neoplasm of this organ—pancreatic ductal carcinoma (PDAC)—is characterized by constitutive activation of STAT3 [69] with a 5-year survival rate <5% [70]. Moreover, STAT3 determines the development of features related to the malignancy of pancreatic cancer. In vivo studies show that blocking STAT3 phosphorylation inhibits the growth of PDAC [69]. As shown in previous studies, WP1066 inhibits proliferation and induces apoptosis of pancreatic cancer cells, reduces the expression of STAT3-dependent anti-apoptotic proteins (Bcl-xL, survivin), and blocks constitutive and IL-6-induced STAT3 phosphorylation [69]. Preliminary in vivo studies show that WP1066 significantly reduces this tumor’s growth rate [70]. Therefore, it is possible to achieve a dual effect of WP1066 supporting pancreatic cancer therapy: inhibition of pSTAT3 activity may limit the induction of damage in skeletal muscle cells and, at the same time, reduce pSTAT3 activity in pancreatic cancer cells and effectively eliminate them.

On the other hand, preclinical studies have shown that WP1066 stimulates the natural immune response to tumors while inhibiting oncogenic transcription factors such as STAT3, but also HIF1-α and c-Myc [71]. As a result, WP1066 belongs to the group of drugs known as “immunity and transduction modulators,” representing excellent potential in clinical oncology.

Another drug candidate STAT3 inhibitor is OPB-31121. OPB-31121 inhibits the phosphorylation of STAT3 and STAT5 without upstream kinase inhibition and has beneficial effects in various types of hematopoietic malignancies, such as multiple myeloma; Burkitt lymphoma; and leukemia harboring BCR-ABL, FLT3/ITD, and JAK2 V61F oncokinases with their oncogenicities dependent on STAT3/5 [72]. OPB-31121 has been shown to bind exclusively to the SH2 domain of STAT3, which provides specific binding with site-directed mutagenesis of critical residues in the SH2 STAT3 domain [72]. Moreover, due to the lack of STAT upstream kinases inhibition, OPB-31121 was safe for normal human blood cells, making it a promising antitumor compound [73]. Two clinical trials concerning OPB-31121 in the treatment of solid tumors and hepatocellular carcinoma have been completed, which gives hope for the further development of this compound as a drug used in clinical oncology.

Pyrimethamine is one of the folic acid antagonists, inhibiting dihydrofolate reductase [74]. Pyrimethamine is a successfully used malarial drug and is undergoing clinical trials to treat chronic lymphocytic leukemia and small lymphocytic leukemia [75]. More interestingly, cachexia–anorexia syndrome is often a phenomenon in the course of malaria [76]. Thus, pyrimethamine has been studied only in the context of antiprotozoal properties, and it may turn out that this compound also protects against weight loss during malaria [77].

Reports on OPB-51602, another oral small-molecule STAT3 inhibitor with potential antineoplastic activity, show its effectiveness in patients with advances solid tumors [78]. OPB-51602 inhibits phosphorylation and thus the activation of STAT3. OPB-51602 interferes with mitochondrial activity, and protein tumor cells expressing a mitochondrially restricted form of STAT3 are highly sensitive to OPB-51602, while STAT3-null cells are protected [79]. Targeting the mitochondrial function of STAT3 induces mortality homeostasis, leading to a synthetic lethality effect in glucose-depleted cancers cells [80] by inhibiting complex I, which could be used in cancer chemotherapy [80]. Regardless of STAT3 inhibition, OPB-51602 also inhibits the mitochondrial respiratory chain and induces a significant increase in mitochondrial superoxide (O2) production (within 1 h of exposure) in the NSCLC H1975 cell line [81]. Further work on OPB-51602 showed an increased dependence on mitochondrial oxidative phosphorylation (XPHOS) in oncogene-dependent tumors exhibiting acquired resistance to targeted therapies. Thus, OPB-51602 gives hope in treating patients who have developed a secondary resistance to standard chemotherapeutic agents such as tyrosine reductase inhibitors, contributing to the reduction of tumor size.

TTI-101 is an example of a STAT3 inhibitor, particularly interesting in terms of muscle cachexia. The use of TTI-101 has been so far explored in muscle loss associated with chronic kidney disease. TTI-101 blocks the STAT3/CCAAT enhancer-binding protein γ, which directly inhibits the myostatin signaling pathway—the one responsible for muscle protein degradation in both chronic kidney disease and cancer [82]. Recruitment for clinical trials on TTI-101 is currently underway in breast cancer, head, neck squamous cell carcinoma, non–small-cell lung cancer, hepatocellular cancer, colorectal cancer, gastric adenocarcinoma, and melanoma, which gives hope for a positive development of this drug.

Napabucasin (BBI608) is an orally bioavailable small molecule known to inhibit cancer stem cells’ activity in the JAK2/STAT3 pathway [83]. Constitutive expression of STAT3 in cancer stem cells independent of upstream signaling regulators confirms the validity of targeting STAT3 in those cells [84]. The positive effects of napabucasin have been confirmed in in vitro, in vivo studies, and clinical trials. Napabucasin reduces the viability of cancer cells and inhibits the renewal of cancer stem cells, and most interestingly, it reduces the pool of cancer stem cells, while standard chemotherapy may increase a sub-population of this type of cell [85]. In clinical trials, napabucasin has been tested both as monotherapy and in combination with cytostatics, mainly with paclitaxel [84]. Positive results of napabucasin action have been confirmed in non–small-cell lung cancer, gastric and gastroesophageal junction (GEJ) adenocarcinoma, bladder cancer, melanoma, ovarian cancer, small-cell lung cancer, esophageal squamous cell cancer, colorectal cancer, and penile squamous cell cancer [85]. Of those mentioned above, the synthetic STAT3 inhibitor napabucasin seems to be the most promising compound, due to its high efficiency and very low toxicity, mainly due to a higher affinity for cancer than normal cells [83]. The clinical development of the selected STAT3 inhibitors is summarized in Table 2.

## 7. Natural STAT3 Inhibitors

It is worth noting that there are also available natural pSTAT3 inhibitors such as CAPE (caffeic acid phenethyl ester), which is isolated from propolis. Numerous in vitro and in vivo reports indicate CAPE’s high efficiency in the induction of apoptosis, characterized by dysregulation of mitochondria and activation of caspase 3 and 7 in cancer cells [86]. Breast cancer reports [87] show that CAPE induces cell cycle arrest, apoptosis, and reduction of expression of transcription factors such as NF-κβ. The last feature may have a direct impact on the inhibition of muscle cachexia. In turn, in vivo studies on the orthotopic model of pancreatic cancer, in the course of which the strongest muscle cachexia is observed, confirm that CAPE inhibits tumor growth [88] while possessing very low cytotoxicity compared to standard chemotherapeutics. CAPE also protects skeletal muscle cells by stimulating glucose uptake and activation of the AMPK pathway (AMP-activated protein kinase) [89]. Other reports mention anti-inflammatory, antioxidant, and protective effects against free radicals in skeletal muscles in the pathophysiology of ischemia–reperfusion (I/R) injury [90].

Capsaicin supplementation is an excellent example of dietary management that inhibits solid tumors [91]. In vivo studies on Walker 256 tumor-bearing rats showed that supplementation with capsaicin (5 mg/kg) reduced tumor size by 49% and reversed serum triacylglycerol concentrations [92]. Moreover, in vivo studies confirm the anticancer effect of capsaicin in skin, prostate, colon, lung, and tongue cancer [93]. However, more and more studies show that constant supplementation of capsaicin increases the feeling of satiety and reduces food intake, which makes the use of this compound in oncological cachexia questionable [93].

Probably the most popular naturally occurring substance is curcumin, a natural diphenolic compound. Curcumin is a well-known modulator of intracellular signaling pathways that govern cancer cell growth, inflammation, and metastasis, thus showing high anticancer properties [94]. The use of curcumin’s anticancer properties as a STAT3 inhibitor has been demonstrated in numerous studies. The positive effect of inhibition of JAK1,2/STAT3 signaling by curcumin has been both confirmed by in vitro studies on multiple myeloma cells [95], cancer stem cells [96], Hodgkin’s lymphoma cells [97], small lung cancer cells [98], and in in vivo models such as glioma development in a syngeneic mouse [99], a rodent model of lung cancer [100] and human non–small-cell lung cancer xenografts [101]. There is no doubt about curcumin’s anticancer properties, but curcumin’s potential beneficial effect on muscle cachexia is highly debatable. In vitro and in vivo experiments are contradictory. Some reports showed a curcumin protective effect against loss of muscle and adipose tissue in oncological cachexia. For example, curcumin administration to tumor-bearing rats did not result in any changes in muscle loss [102,103] or even a negative effect on weight loss, as demonstrated in studies with curcumin supplementation in advanced pancreatic cancer patients [104]. Other reports indicated a beneficial effect of curcumin on muscle cachexia [105,106]. However, further research on curcumin’s influence on oncological cachexia is needed [107]. Among other natural STAT3 inhibitors with antitumor properties, butein [108] and ursolic acid [109] are worth mentioning. Therefore, dietary management involving supplementation of natural STAT3 inhibitors may help prevent alleviating changes during the development of cachexia in skeletal muscles.

## 8. Perspectives

Based on the described above rationale and the critical role of STAT3 in oncogenesis and muscle cachexia processes, we hypothesize that STAT3 blockage could exert a dualistic effect—anticancer cytotoxicity, and at the same time, disrupt the cachectic signaling pathway. Our hypothesis is currently under investigation in our laboratory. The schematic representation of the hypothesis mentioned above is illustrated in Figure 2. As far as we know, no available studies evaluate the biological effects of STAT3 inhibition on cancer-induced cachexia in one experimental model.

## Figures and Tables

**Figure 1 ijms-21-08261-f001:**
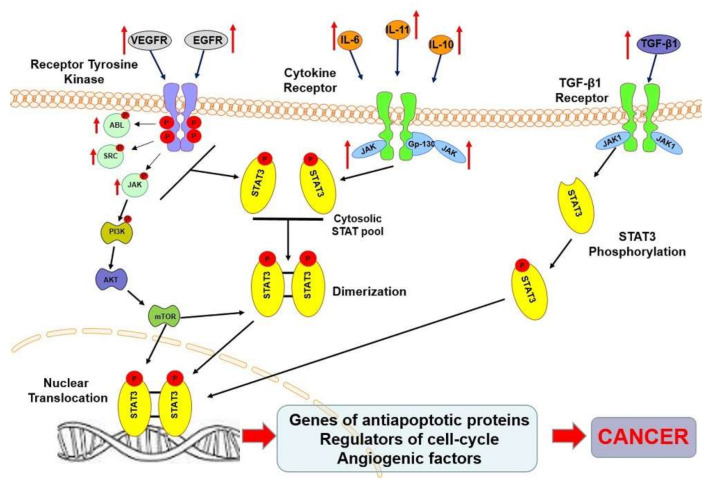
Schematic representation of the signaling pathways leading to signal transducer and activator of transcription 3 (STAT3) activation. Increased levels (indicated by red arrows) of IL-6, IL-10, IL-11, VEGF, EGFR, TGF-β, JAK, Abl, or Src lead to an increase in the rate of phosphorylated form of STAT3 protein (marked with P)-P-STAT3. Moreover, increasing the intracellular JAK level activates the mTOR pathway, which can increase the level of STAT3 dimerization both in the cytoplasm and in the nucleus. Upregulated pSTAT3 levels can lead to increased expression of genes encoding anti-apoptotic proteins, cell cycle regulators, or angiogenic factors. All of this, as a consequence, can induce the formation of neoplastic cells [22,23,25,26].

**Figure 2 ijms-21-08261-f002:**
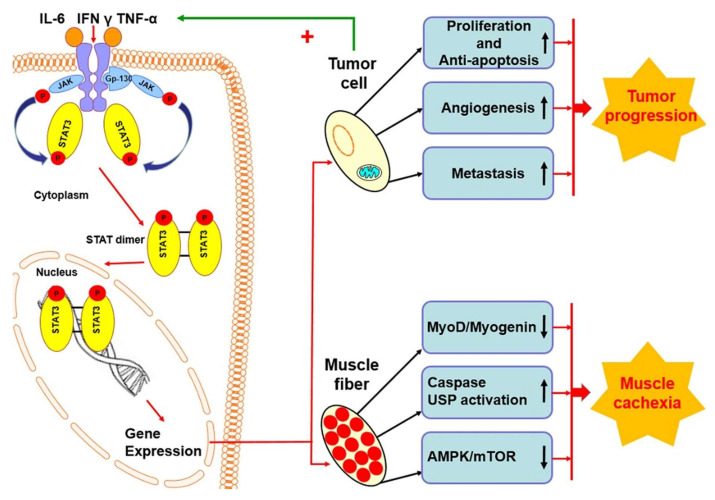
Comparison of the consequences of excessive STAT3 activation in muscle and cancer cells. STAT3 hyperactivity in myocytes leads to an increase (marked with an up arrow) in caspase activity, leading to universal stress proteins’ (USP) involvement. In the area of muscle tissue, a decrease in myogenin and MyoD expression (marked with a down arrow), and a weakening of the AMP-activated protein kinase (AMPK)/mTOR signal pathway were observed, which induces the process of muscle cachexia. Neoplastic cells increase the level (marked with an up) of angiogenesis, metastasis, proliferation, and inhibition of apoptosis. In this case, these events lead to the progression of the neoplastic process. On the other hand, tumor cells secrete pro-inflammatory cytokines, including IL-6, which additionally stimulate the activation of STAT3 (marked with +), intensifying pathological changes in skeletal muscles and stimulating tumor progression.

**Table 1 ijms-21-08261-t001:** Current clinical trials on muscle cachexia treatment.

Compound	Mechanism of Action	Indication	Clinical Trial ID	Chemical Structure
**Anamorelin hydrochloride**	selective agonist of the ghrelin/growth hormone secretagogue receptor	cancer cachexia, non–small-cell lung cancer (NSCLC)	NCT03743064, NCT03637816, NCT03743051, NCT01387269, NCT01387282, NCT03035409, NCT01395914, NCT00622193	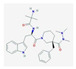 PubChem Identifier: CID 9828911, https://pubchem.ncbi.nlm.nih.gov/compound/Anamorelin
**Relamorelin (RM-131)**	selective agonist of the ghrelin/growth hormone secretagogue receptor	anorexia nervosa	NCT01642550	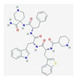 PubChem Identifier: CID 85364156, https://pubchem.ncbi.nlm.nih.gov/compound/Relamorelin
**NGM120** **Monoclonal antibody against GDNP protein alpha-like receptor (GFRAL)–3P10 antibody**	GDNF family receptor-α-like (GFRAL)-Ret proto-oncogene (RET) blocker	cancer cachexia	NCT04068896	
**Vitamin D**	promotion of lipid partitioning and muscle metabolic function	cancer cachexia	NCT03144128	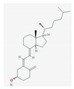 PubChem Identifier: CID 5280795, https://pubchem.ncbi.nlm.nih.gov/compound/Cholecalciferol
**Branched Chain Amino Acid (BCAA)**	regulation of the anabolic pathway of muscle synthesis	sarcopenia in chronic liver disease	NCT04246918	
**Omega-3 fatty acids**	regulation of cell signaling, cell structure, and fluidity of membranes	cancer cachexia	NCT01596933, NCT00031707	
**Beta-hydroxy-beta-methyl butyrate (HMB)**	improvement of muscle hypertrophy and strength, aerobic performance, resistance to fatigue, and regenerative capacity	critical illness, cancer cachexia	NCT03464708, NCT03151291	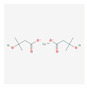 PubChem Identifier: CID 9860341, https://pubchem.ncbi.nlm.nih.gov/compound/Calcium-beta-hydroxy-beta-methylbutyrate

**Table 2 ijms-21-08261-t002:** Current clinical trials of STAT3 inhibitors.

Compound	Mechanism of Action	Indication	Clinical trial ID	Chemical Structure
**WP1066**	cell-permeable JAK2, STAT3, STAT5, and ERK1/2 inhibitor, responsible for the dephosphorylation and nuclear export of constitutively phosphorylated STAT3	metastatic malignant neoplasms in the brain; metastatic melanoma; recurrent glioblastoma; recurrent brain neoplasm	NCT04334863 NCT01904123	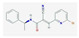 PubChem Identifier: CID 11210478, https://pubchem.ncbi.nlm.nih.gov/compound/wp1066
**OPB-31121**	potent inhibition of STAT3 and STAT5 phosphorylation without upstream kinase inhibition	advanced cancer, solid tumors, hepatocellular carcinoma	NCT00955812 NCT01406574	
**TTI-101**	binaphthol sulfonamide-based inhibitor of STAT3 that specifically targets and binds to the phosphotyrosine peptide-binding site within the Src homology 2 (SH2) domain of STAT3	breast cancer, head, and neck squamous cell carcinoma, non–small-cell lung cancer, colorectal cancer, gastric adenocarcinoma, melanoma	NCT04068896	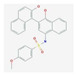 PubChem Identifier: SID 382371065, 432001-19-9, https://pubchem.ncbi.nlm.nih.gov/substance/382371065
**Pyrimethamine**	synthetic derivative of ethyl-pyrimidine, a competitive inhibitor of dihydrofolate reductase (DHFR)—a key enzyme in the redox cycle for tetrahydrofolate production; a cofactor required for DNA and proteins synthesis	relapsed chronic lymphocytic leukemia, small lymphocytic lymphoma	NCT01066663	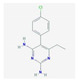 PubChem Identifier: CID 4993, https://pubchem.ncbi.nlm.nih.gov/compound/Pyrimethamine
**OPB-51602**	inhibition of STAT3 phosphorylation and activation of STAT3	advanced solid tumors: breast cancer, head, and neck squamous cell carcinoma, non–small-cell lung cancer, hepatocellular cancer, colorectal cancer, gastric adenocarcinoma, melanoma	NCT01423903 NCT01344876 NCT01184807	
**Napabucasin (Napa, BBI608)**	STAT3 and cancer cell stemness inhibitor	gastrointestinal malignancies, pancreatic cancer, GBM	NCT03721744 NCT02753127 NCT03522649	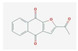 PubChem Identifier: CID 10331844 https://pubchem.ncbi.nlm.nih.gov/compound/Napabucasin

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
