# Peer review of "Targeting the JAK2/STAT3 Pathway—Can We Compare It to the Two Faces of the God Janus?"

_ijms, 2020, doi:10.3390/ijms21218261_

Round 1
Reviewer 1 Report
The review article is well written and there is ample supporting evidence for the role of STAT3 in Muscle cachexia. The efforts taken by the authors to present this review is appreciated.
I recommend rechecking the current literatures for the most up-to-date information and modify references wherever possible for during the final proof reading of the review article.
I recommend acceptance of the review article for publication.
Author Response
We acknowledge the reviewers' comments, who very accurately indicated the weak points of our paper. All comments were taken into account, and the changes visibly enriched the content of the paper. We hope that the reviewers will be satisfied with the current form of the manuscript.
Review 1
Comments and Suggestions for Authors
The review article is well written, and there is ample supporting evidence for the role of STAT3 in Muscle cachexia. The efforts taken by the authors to present this review is appreciated. I recommend rechecking the current literatures for the most up-to-date information and modify references wherever possible for during the final proof reading of the review article.
- The references were corrected, and current articles have replaced 17 papers published before 2000 year. All cited references were published up to 2000 year, and 81/111 references were published during the last ten years.
Reviewer 2 Report
The manuscript highlights the role of STAT3 in cachexia as well as current inhibitors/therapies that could potentially be translated into treating STAT3-driven muscle atrophy. The topic is very important and the manuscript was well organized, easy to follow, and will be suitable for publication.
There are a few critical points that need to be resolved and some additional references should be included. See the following points below:
- There is confusion with the term “STAT3 kinase” which is heavily used within the manuscript. This is a critical point that needs to be rectified before the manuscript can be published: Does “STAT3 kinase” refer to STAT3 or JAK2? If the former case, then this is incorrect since the protein STAT3 is not a kinase and does not have any known phosphorylation activity or any catalytic domains for kinase activity. If the latter, please replace the term STAT3 kinase with JAK2 for clarity in all cases. Also note there are multiple kinases for STAT3, which is why this term is not generally used.
- Abstract: Change “STAT-3” to “STAT3”
- Page 2: “It should be aware that oncological disease is associated with suppression of appetite” should be re-worded
- Page 4: The role of STAT3 in skeletal muscle regeneration is interesting. Some more information seems pertinent in resolving the function of STAT3. For instance, elimination of STAT3 results in impairment of muscle cell differentiation, but it is also stated that transient STAT3 blockade results in increased muscle regeneration. STAT3 is a valid target in DMD, and I believe both of these statements are true, but some more information on distinguishing when STAT3 results in simulation/impeding muscle generation would be of interest to the reader. Also, knockout of STAT3 usually results in embryonic lethality, so a few more statements on the type of STAT3 knockout mice seem appropriate in this section; Alternatively, this section can be slightly expanded or also discussed in the “Perspectives”, since it really supports the title of the manuscript, with STAT3 have contrasting roles in muscle development.
- Page 5: Spelling “cuntless” to “countless”
- Page 6” which makes them great hope in cancer research” should be re-worded
- Page 6 completed, which gives hope for further development as a drug of that molecule.” should be reworded
- Structures for compounds should be provided where available (WP, pyrimethamine, OBP etc.)
- Page 7” treatment o patients” to “treatment of patients”
- Napabucasin is described in Table 1 but not discussed in the text
- Figure 1 seems incomplete, a reference to Fig 1 of a recent review would seem appropriate highlighting the additional pleiotropic elements within the STAT pathways: 10.3390/cancers12082002
- The follow references also seem appropriate to highlight:
10.3390/ijms19082265
10.1097/MCO.0000000000000273
Fig 1. Also note the STAT3 phosphorylated dimers should be “head-to-head” instead of “head-to-tail” (SH2 domains dimerize head-to-tail, but the full STAT proteins dimerize “head-to-head”). Fig 2, Correct the spelling of Myogenin, same STAT dimer issues as above.
Author Response
We acknowledge the reviewers' comments, who very accurately indicated the weak points of our paper. All comments were taken into account, and the changes visibly enriched the content of the paper. We hope that the reviewers will be satisfied with the current form of the manuscript.
Review 2
Comments and Suggestions for Authors
- There is confusion with the term “STAT3 kinase” which is heavily used within the manuscript. This is a critical point that needs to be rectified before the manuscript can be published: Does “STAT3 kinase” refer to STAT3 or JAK2? If the former case, then this is incorrect since the protein STAT3 is not a kinase and does not have any known phosphorylation activity or any catalytic domains for kinase activity. If the latter, please replace the term STAT3 kinase with JAK2 for clarity in all cases. Also note there are multiple kinases for STAT3, which is why this term is not generally used.
- We agree with the Reviewers that we used the misleading "STAT3 kinase" term. Whenever STAT3 was mentioned, it refers to STAT3 protein, not upstream kinases. The manuscript has been corrected, and "STAT3 kinase" has been corrected to "STAT3 protein".
- Abstract: Change “STAT-3” to “STAT3”
- It was corrected according to the Reviewer’s suggestion.
- Page 2: “It should be aware that oncological disease is associated with suppression of appetite” should be re-worded
- It was corrected according to the Reviewer’s suggestion.
- Page 4: The role of STAT3 in skeletal muscle regeneration is interesting. Some more information seems pertinent in resolving the function of STAT3. For instance, elimination of STAT3 results in impairment of muscle cell differentiation, but it is also stated that transient STAT3 blockade results in increased muscle regeneration. STAT3 is a valid target in DMD, and I believe both of these statements are true, but some more information on distinguishing when STAT3 results in simulation/impeding muscle generation would be of interest to the reader. Also, knockout of STAT3 usually results in embryonic lethality, so a few more statements on the type of STAT3 knockout mice seem appropriate in this section; Alternatively, this section can be slightly expanded or also discussed in the “Perspectives”, since it really supports the title of the manuscript, with STAT3 have contrasting roles in muscle development.
- We agree with the Reviewer's comments, and additional information about the role of STAT3 in early muscle development has been added.
- Page 5: Spelling “cuntless” to “countless”
- It was corrected according to the Reviewer’s suggestion.
- Page 6” which makes them great hope in cancer research” should be re-worded
- It was corrected according to the Reviewer’s suggestion.
- Page 6 completed, which gives hope for further development as a drug of that molecule.” should be reworded
- It was corrected according to the Reviewer’s suggestion.
- Structures for compounds should be provided where available (WP, pyrimethamine, OBP etc.)
- Available in PubChem database (allowed for publication reuse without any special permission), chemical structures of STAT3 inhibitors used in clinical trials have been added in Tables.
- Page 7” treatment o patients” to “treatment of patients”
- It was corrected according to the Reviewer’s suggestion.
- Napabucasin is described in Table 1 but not discussed in the text
- Additional information concerning Napabucasin has been added to the main text.
- Figure 1 seems incomplete, a reference to Fig 1 of a recent review would seem appropriate highlighting the additional pleiotropic elements within the STAT pathways: 10.3390/cancers12082002
The follow references also seem appropriate to highlight:10.3390/ijms19082265; 10.1097/MCO.0000000000000273
- Figure 1 has been corrected. Additional pleiotropic elements have been added. Also mentioned above references have been included.
- Fig 1. Also note the STAT3 phosphorylated dimers should be “head-to-head” instead of “head-to-tail” (SH2 domains dimerize head-to-tail, but the full STAT proteins dimerize “head-to-head”).
- Figure 1 has been corrected.
- Fig 2, Correct the spelling of Myogenin, same STAT dimer issues as above.
- Figure 2 has been corrected.
We have done our best to improve the English.
Round 2
Reviewer 2 Report
All changes have been addressed, and the review is a lot stronger and a nice addition to the field.